# A Meta-Analysis to Estimate Prevalence of Resistance to Tetracyclines and Third Generation Cephalosporins in Enterobacteriaceae Isolated from Food Crops

**DOI:** 10.3390/antibiotics11101424

**Published:** 2022-10-17

**Authors:** Ariel A. Brunn, Manon Roustit, Zaharat Kadri-Alabi, Luca Guardabassi, Jeff Waage

**Affiliations:** 1Department of Public Health, Environment and Society, London School of Hygiene and Tropical Medicine, London WC1E 7HT, UK; 2Department of Pathobiology and Population Sciences, Royal Veterinary College, London NW1 0TU, UK; 3Department of Veterinary and Animal Sciences, University of Copenhagen, 1165 Copenhagen, Denmark; 4Department of Disease Control, London School of Hygiene and Tropical Medicine, London WC1E 7HT, UK

**Keywords:** antimicrobial resistance, risk assessment, food value chains, food crops, agriculture

## Abstract

Application of human and animal waste to fields and water sources and on-farm antimicrobial usage are documented contributors to the occurrence of antimicrobial resistance (AMR) in agricultural domains. This meta-analysis aimed to determine the prevalence of resistance to tetracycline (TET) and third generation cephalosporins (3GC) in Enterobacteriaceae isolated from food crops. TET was selected in view of its wide use in agriculture, whereas 3GC were selected because of the public health concerns of reported resistance to these critically important antibiotics in the environment. Forty-two studies from all six world regions published between 2010 and 2022 met the eligibility criteria. A random effects model estimated that 4.63% (95% CI: 2.57%, 7.18%; *p*-value: <0.0001) and 3.75% (95%CI: 2.13%, 5.74%; *p*-value: <0.0001) of surveyed food crops harboured Enterobacteriaceae resistant to TET and 3GC, respectively. No significant differences were observed between pre- and post-harvest stages of the value chain. 3GC resistance prevalence estimates in food crops were highest for the African region (6.59%; 95% CI: 2.41%, 12.40%; *p*-value: <0.0001) and lowest for Europe (1.84%; 95% CI: 0.00%, 6.02%; *p*-value: <0.0001). Considering the rare use of 3GC in agriculture, these results support its inclusion for AMR surveillance in food crops. Integrating food crops into One Health AMR surveillance using harmonized sampling methods could confirm trends highlighted here.

## 1. Introduction

Few global health problems are as pervasive and insidious as antimicrobial resistance (AMR). That bacteria employ various mechanisms to withstand the effects of antibiotics has long been understood by biomedical researchers [1]. However, the accelerated process of acquisition of resistance traits, now extensively documented in a diversity of microbes and geographies, has elevated awareness of this growing threat to health to the global stage [2,3]. Though AMR research and policy efforts have centred on medical and livestock sectors, the wider environment is under renewed scrutiny as a conduit between human and animal reservoirs, with accumulating evidence of AMR presence in watersheds and croplands [4,5,6]. Empirical data suggests that direct or indirect (e.g., through contaminated irrigation water) application of livestock and human wastes onto agricultural fields inadvertently contaminates soils and crops with resistant microbes; antimicrobial and biocidal crop treatments are under-reported and are plausible amplifiers for resistance emergence in the crop microbiome [7,8,9]. Since consumption trends indicate a growing preference for plant-based diets which are frequently eaten with minimal preparation or heat treatment, food crops may be an important AMR hazard to consumers and food value chain actors [10].

The health repercussions of AMR ingestion, however, are weakly understood and risk assessments are hampered by a lack of clinical and epidemiological data [11,12]. National statistics of AMR in food crops are rare and frequently based on temporary surveillance programs or limited surveys [13,14]. Moreover, the absence of harmonised standards currently impedes international monitoring efforts despite pressure from consumer groups for improved transparency within increasingly complex and interconnected food value chains [15]. Major crop growing and exporting regions also lack incentives to participate in voluntary schemes, the results of which might compromise trade relationships with importing countries.

In lieu of national datasets, an increasing number of observational studies conducted in fields and markets report the presence of resistant microbes on food crops [16]. Global and regional prevalence estimates of AMR contamination in plant-based foods could facilitate AMR policymaking efforts by contributing to exposure assessments already being derived for animal-source foods [17,18]. Defining the risk of ingestion of AMR in plant-based foods would allow a more complete understanding of consumption and occupational exposure risks in agricultural and food-producing sectors.

AMR contamination of food crops occurs before harvest (pre-harvest stages) as a consequence of contaminated inputs such as manure, irrigation water or crop treatments. Following harvest, as produce enters the processing chain (Figure 1), cross-contamination can occur through equipment or handling from value chain actors, such as the farm labour force, referred to as post-harvest contamination [19]. Transfer of acquired antimicrobial resistant bacteria (ARB) from agricultural inputs, labourers, or equipment is considered a more probable pathway than de novo emergence in the plant microbiome [12,20].

Use of meta-analysis methods to estimate pooled prevalence from existing literature has had limited uptake in food safety studies in produce [21,22,23,24,25]. However, AMR within the food crop value chain is receiving renewed attention and greater efforts being made to refine quantitative microbial risk assessments with available sources of data [26,27,28]. Through the application of random effects models, this review employs point prevalence estimates of AMR in Enterobacteriaceae isolated from food crops extracted from observational studies to generate baseline global prevalence estimates. This study focused on Enterobacteriaceae as these bacteria are opportunistic pathogens that are commonly used as indicators in AMR surveillance studies. Selection of AMR results was limited to tetracycline (TET) antibiotics due to their widespread use in crop and veterinary applications and abundance in livestock waste [5], and third generation cephalosporin (3GC) antibiotics which are classified by the World Health Organization (WHO) as Highest Priority, Critically Important Antimicrobials [29]; their use in agriculture has been repeatedly called into question due to rising concerns of globally disseminated resistance [30,31].

## 2. Results

There were 8517 studies overall retrieved from four bibliographic databases that had been published between 1 January 2000, and 5 April 2022, to include a total of forty-two studies in the meta-analysis (Figure 2). Prevalence estimates from these studies were calculated from 149,298 samples that met the meta-analysis inclusion criteria on AMR in Enterobacteriaceae isolated from food crops.

All included studies were observational in design. Somewhat fewer prevalence estimates were extracted for phenotypic 3GC resistance (*n* = 27), than for TET resistance (*n* = 30) and the post-harvest stage of the value chain was the more common stage for sampling (*n* = 43; pre-harvest stage: *n* = 18). Studies originated from 34 countries including all six world regions but favoured the Western Pacific Region (*n* = 12), Europe (*n* = 11), and Africa (*n* = 8), with the fewest studies conducted in South East Asia (*n* = 7), the Americas (*n* = 4) and the Eastern Mediterranean region (*n* = 4).

### 2.1. Risk of Bias Assessment

Most studies (*n* = 18) graded for bias using the AXIS Risk of Bias tool [33] were found to have a high risk of bias due to methodological constraints, while very high and moderate risks of bias were observed in nine and fifteen studies, respectively, (Table 1). No studies were determined to have a low risk of bias.

### 2.2. Subgroup Meta-Analysis

#### 2.2.1. Enterobacteriaceae Prevalence in Food Crops

Data was extracted from 42 studies that reported Enterobacteriaceae isolation from food crops using culture-based methods to yield an overall prevalence of 11.45% (95% CI: 7.96%, 15.45%; *p*-value: <0.0001). Prevalence estimates were disaggregated by value chain stage of sampling and indicate a slightly higher prevalence of 13.86% (95% CI: 5.78%, 24.53%; *p*-value: <0.0001) calculated for samples obtained at the pre-harvest stage in comparison to 10.85% (95% CI: 7.30, 14.97%; *p*-value: <0.0001) for samples selected in post-harvest settings (Table 2).

#### 2.2.2. Total AMR Prevalence in Produce

Meta-analysis of 3GC and TET prevalence estimates from all stages of the food value chain based on 45 prevalence estimates extracted from the available literature yielded a global pooled prevalence of 4.75% (95% CI: 2.92%, 6.94%; *p*-value: <0.0001) of 149,298 food crop samples that were tested.

#### 2.2.3. AMR Prevalence by Antimicrobial Class

Prevalence estimates of 3GC and TET were meta-analysed by subgrouping of antimicrobial class. In total, 27 prevalence estimates were extracted for 3GC resistance detected in food crops with an overall prevalence of 3.75% (95% CI: 2.13%, 5.74%; *p*-value: <0.0001). A combined prevalence estimate for TET resistance in food crops was 4.63% (95% CI: 2.57%, 7.18%; *p*-value: <0.0001), based on 30 estimates.

#### 2.2.4. AMR Prevalence by Stage of Value Chain Sampling

The effect on prevalence of sampling at different stages of the food crop value chain was assessed after subgrouping by antimicrobial class. A pooled prevalence statistic for 3GC resistance of 4.45% (95% CI: 1.44%, 8.71%; *p*-value: 0.0031) was comparable to the pooled prevalence of TET resistance of 4.55% (95% CI:1.83%; 8.24%; *p*-value: <0.0001) for samples collected in farms and gardens prior to harvest and processing. In samples collected within the post-harvest stages of the value chain, the 3GC resistance prevalence declined to 3.44% (95% CI: 1.64%, 5.76%; *p*-value: <0.0001), whereas a slight increase to 4.72% (95% CI: 2.11%, 8.16%; *p*-value: <0.0001) was observed for TET resistance.

#### 2.2.5. AMR Prevalence by Region and Antimicrobial Class Type

Further subgroup analysis was conducted by disaggregating estimates by AMR resistance type and region. Pooled prevalence estimates were calculated for the WHO health regions of the Americas, Africa, South East Asia, Western Pacific, Eastern Mediterranean, and Europe combining the sampling stages together. Pooled prevalence estimates of TET resistance varied from 2.48% for the Americas (*n* = 4; 95% CI: 0.00%, 8.82%; *p*-value: <0.0001) to 6.28% for Europe (*n* = 6; 95% CI: 0.68%, 15.72%; *p*-value: <0.0001) and for 3GC resistance, from 1.84% for Europe (*n* = 5, 95% CI: 0.00%, 6.02%; *p*-value: <0.0001) to 6.59% for Africa (*n* = 6; 95% CI: 2.41%, 12.40%; *p*-value: <0.0001) (Table 3).

Combining AMR types resulted in the lowest prevalence of 1.46% (95% CI: 0.36%, 3.13%; *p*-value: 0.0002) from the Eastern Mediterranean region and 5.44% in the African region (95% CI: 2.54%, 9.24%; *p*-value: <0.0001). Results were not meta-analysed for AMR classes for regions from which fewer than four studies were available.

### 2.3. Meta-Regression

To assess the impact of between-study heterogeneity, univariable meta-regression methods were used. Trial sizes of <100 samples tested are a possible source of heterogeneity between studies (*p*-value: 0.0019), however WHO health region, country income status, value chain stage of sampling, sample weight, risk of bias classification, or Enterobacteriaceae genus were not significant sources of heterogeneity.

### 2.4. Sensitivity Analysis

Prevalence estimates from studies that were assessed to have a very high risk of bias or that tested fewer than 100 samples were removed in the sensitivity analysis to examine the influence of less robust methods of data collection on the pooled prevalence value. The removal of these studies yielded a new global prevalence estimate of 3.09% (95% CI: 1.73%, 4.81%; *p*-value: <0.01), which lies within the confidence interval of the original global pooled prevalence. Additional sensitivity analysis was conducted to determine the effect of ESBL-selection methods on AST prevalence. Studies which first selected ESBL-producing isolates had a lower pooled prevalence of TET resistance and higher 3GC resistance than studies that did not use these methods (Appendix B).

### 2.5. Publication Bias

Publication bias was assessed by constructing a funnel plot (Figure 3) and assessing tests for bias. The funnel plot showed a right-sided skew of prevalence estimates; this asymmetry was confirmed by significant *p*-values for the Egger’s regression test (*p*-value: <0.0001) and Thompson linear regression test (*p*-value: <0.0001).

### 2.6. AMR Prevalence Ratio

The prevalence ratio was calculated to be 42.7% and indicates the relative proportion of 3GC- and TET-positive samples to Enterobacteriaceae-positive samples amongst food crop samples.

## 3. Discussion

Using meta-analysis methods and the available literature, this study provides a baseline estimate that 4.75% (95% CI: 2.92%, 6.94%) of food crops surveyed globally for AMR are contaminated with TET- or 3GC-resistant Enterobacteriaceae. As this is based on point prevalence estimates from sub-national observational surveys, this estimate should be carefully interpreted as the potential scale of resistance in food crops for these two antimicrobial classes. The food safety sector has thus far been preoccupied with animal-derived foods to characterize the AMR hazard associated with veterinary use of antibiotics [17,18], however the authors contend that plant-based foods are an overlooked hazard of consumption and occupational AMR exposures. Although the prevalence estimates for TET (4.63%) and 3GC (3.75%) were similar, the estimated occurrence of 3GC resistance is of particular concern due to the clinical importance of third generation cephalosporins, which are classified as “highest priority, critically important” by the WHO [29] and category B (“Restrict”) by the European Medicine Agency [75] whereas tetracyclines are ranked as “highly important” and category D (“Prudence”), respectively. This result is surprising considering the widespread use of TET in agriculture, which is in contrast with limited use of 3GCs outside human clinical settings and indicates a need for inclusion of food crops in AMR surveillance activities at national levels. Consumer trends towards plant-based diets which are frequently eaten in a raw form or with minimal preparation further necessitate enhanced surveillance on this potentially important source of community AMR acquisition and spread.

Resistance to 3GCs represent a significant AMR challenge relevant to human health and ESBL-producing *E. coli* are a proposed indicator for development of National Integrated Surveillance Systems for AMR in the WHO Global Antimicrobial Resistance Surveillance System (GLASS) [76]. This standardised surveillance scheme, currently piloted in several countries, uses the One Health concept to develop a multisectoral sentinel program to gather evidence of AMR dissemination characteristics. Despite its ambitions, the sampling protocol does not include food crops, relying instead on sources of water as the sole environmental sample. In light of our findings of 3GC resistant Enterobacteriaceae in food crops and potential consumer and occupational risks thereof, adaptation of the scheme to include food crop samples would satisfy the objectives for a fully integrated surveillance scheme.

A total of 42 studies provided phenotypic AMR prevalence estimates in Enterobacteriaceae isolated from food crops. Over one-third of studies were published in the 18 months prior to our review, suggesting growing attention from the research community. Included studies predominantly sampled produce in the post-harvest stage of the value chain, likely related to difficulties in collecting samples from privately-owned farms as opposed to publicly accessible markets. The scarcity of farm-level data impacts the robustness of our pre-harvest pooled prevalence estimates and may obscure the true prevalence of AMR contamination at source. Nonetheless, surveillance of food safety and quality in retailed foods is preferable as it contributes data to support implementable risk reduction activities by consumers, such as heating produce before consumption [15,77].

Prevalence estimates derived from studies conducted in the Western Pacific world region occurred most frequently in our review, whereas studies conducted in the American and Eastern Mediterranean regions were underrepresented. In world regions with sufficient studies to perform subgroup analysis by antimicrobial class, a general trend of slightly lower prevalence of AMR for 3GC than for TET is observed, however when world region was evaluated in a meta-regression model, it was not a significant source of heterogeneity. While pooled prevalence estimates from the African region suggest a higher prevalence of 3GC resistance in comparison to TET resistance, conclusions cannot yet be drawn on this pattern. Additional AMR surveillance in the African region might further elucidate the cause of this incongruence.

This study reports a global pooled prevalence of 11.45% contamination of produce with Enterobacteriaceae (95% CI: 7.96%, 15.45%; *p*-value: <0.0001), over twice as high as estimates for enteric pathogens *E. coli* and *Salmonella* spp. in produce reported in the literature (4.1% prevalence in lettuce, range of 2.8–6.4% for high and low development status countries, respectively) [24]. In contrast, our results on AMR prevalence in produce are broadly comparable to a national prevalence estimate published for AMR in food crops in Switzerland [13]. This quantitative analysis of multiple classes of AMR in in foodborne bacteria, indicator bacteria and lactic acid bacteria isolated from meat, dairy, and plant food samples produced or imported into Switzerland for retail reported a prevalence of 2.6% of whole, fresh, plant-based foods had any type of AMR.

Sub-group analysis yielded similar prevalence estimates between pre- and post-harvest stages of the food crop value chain, irrespective of antibiotic class, and value chain stage was not statistically significant in meta-regression analysis. It is possible that the low number of studies on pre-harvest samples are not representative of the true prevalence or that particular laboratory methods alter the sensitivity of detection of resistant isolates. Alternately, it is possible that cross-contamination of AMR isolates has minimal impact on surface contamination and that AMR elements internalised through root uptake or wounded tissue inoculation is the primary driver for food crop contamination [78]. Since the most common microbial isolation methods homogenize plant tissue prior to growth in a bacterial medium, it cannot be ascertained whether surface or tissue-embedded AMR contamination is a more important risk factor for AMR carriage. However, it is possible that plant tissue uptake of AMR elements may be a more considerable source of AMR propagation through the food value chain and may in part explain the null effect of value chain stage of sampling in this study.

A high level of bias was found in most studies included in the meta-analysis. Selection bias, in the form of absent sample size justification and no reported randomisation of sampling, was the most commonly encountered bias. In addition, there was substantial heterogeneity among studies in relation to the methods used for bacterial isolation, identification, and susceptibility testing. Confounding bias and reporting bias were also assessed as high. The high heterogeneity of the pooled prevalence estimates indicated by the I^2^ value reflect the large differences in sample size and prevalence values. Meta-analysis of prevalence has been reported to generate high statistical heterogeneity due to substantial methodological differences [79]. The addition of further studies to increase the combined sample size could improve the internal validity of results.

There were several limitations to our review. Importantly, this study aimed to characterise the prevalence of AMR contamination among plant-based food samples, but does not estimate the “dose” of resistant bacteria present per sample. The latter estimate is critical for dose–response calculations and some dose–response models for ingestion of AMR bacteria have been proposed [80]. However, it is still largely unclear what health risk the consumption of any resistant microbe might cause, and further empirical data is needed.

This meta-analysis was also limited to AMR detected through culture-based methods and did not include prevalence estimates of resistant Enterobactericeae-contaminated food crops detected through PCR or genomic sequencing. Studies reporting antimicrobial resistance genes (ARG) from food crop samples reported their occurrence inconsistently and several authors conducted antimicrobial resistance testing on phenotypically confirmed ESBL isolates, with implications for the prevalence estimate denominator. Though previous authors have recommended that genomic methods be incorporated into quantitative microbial risk assessments [28,81], a lack of standardized testing methods and limited access to equipment and training for low- and middle-income countries currently constrain the utility of published data on food crops for secondary analysis.

Heterogeneity of methods also impacts prevalence estimates. Different types of agricultural production systems, food crop types, agricultural inputs, and retail systems, may be difficult to examine in an epidemiological analysis. For example, almost half of all studies used culture-based methods to select for ESBL-producing bacteria prior to antimicrobial susceptibility testing. These methods are intended to enhance detection of resistance to beta-lactam antibiotics, and a small positive effect was found on 3GC pooled prevalence in studies that conducted antibiotic susceptibility testing on ESBL-producing bacteria. However, TET pooled prevalence estimates were lower among studies that used ESBL-selection methods in comparison to those that did not. For this reason and given the wide use of tetracyclines in the veterinary sector, the authors hypothesize that the pooled prevalence estimates for tetracycline resistance assessed in this study may be an underestimate.

## 4. Materials and Methods

A scoping review of studies reporting AMR in opportunistic and obligate human pathogens isolated from food crops was conducted in 2020 following PRISMA reporting guidelines for Scoping Reviews [82]. MEDLINE, Scopus, CAB Direct and EMBASE bibliographic databases were searched for articles published between 1 January 2000, to 21 July 2020, and yielded 6118 studies, from which data was extracted on 196 studies. The full methods of the scoping review have been previously described [16]. Following the narrative review, a subset of studies providing point prevalence estimates of resistance harboured by Enterobacteriaceae to TET and/or 3GC antibiotics were selected for meta-analysis. These two classes of antibiotics were prioritized as they are used in veterinary and agricultural applications and designated by the World Organization for Animal Health (WOAH) as veterinary critically important antibiotics, in which they are considered indispensable for treatment of certain animal diseases where effective replacement options are not available [83]. An update of all four databases was completed using the original search strings in April 2022.

Studies were included in the meta-analysis if they provided prevalence estimates for either one or both AMR types (TET and 3GC resistance), culture-based AMR detection methods were reported, the total sample size of food crops as well as the number of relevant AMR-positive food crops samples were extractable, and if the stage of the food crop value chain was clearly defined for sample collection. Studies were excluded if they did not report resistance to the two specified antibiotics. To reduce heterogeneity related to methods and interpretation of antimicrobial susceptibility testing, we excluded studies that did not use internationally recognized interpretive criteria for antimicrobial susceptibility testing, such as those defined by the Clinical and Laboratory Standards Institute (CLSI) or the European Committee on Antimicrobial Susceptibility Testing (EUCAST). For the same reason, studies conducted prior to 2010 were also excluded since CLSI breakpoints for 3GC were changed in that year [84]. Two-stage screening and data extraction was done with one author arbitrating on consensus decisions.

The AXIS Risk of Bias Appraisal Tool for Cross-sectional Studies was used to evaluate bias at the study and outcome level [33]. In total, 42 studies were assessed by one researcher (MR) with one-third of the studies assessed by a second researcher (AB) for validation. No studies were excluded based on the result of their bias assessment, rather, the impact of study quality was explored in the sensitivity analysis.

Prevalence estimates were stratified by value chain stage of sampling (pre-, post-harvest or both), antimicrobial resistance type (TET or 3GC or both), and geographical region using the WHO health region classification (South East Asia, Western Pacific, Eastern Mediterranean, Africa, Europe, Americas). Point prevalence estimates were transformed using the Freeman-Tukey double-arcsine transformation and all meta-analysis conducted on the transformed proportions and confidence intervals (CI) using a random-effects model. Back-transformation provided final pooled estimates of prevalence [85]. Subgroup analysis was used to assess heterogeneity of prevalence for antibiotic-class type; value-chain stage; and world region. A meta-regression was undertaken to evaluate the effects of methodological heterogeneity on stage of value chain sampling, world region, country development status, risk of bias, microbe type, sample weight, and trial size.

I^2^ values quantify the level of between-study heterogeneity. A value of I^2^ greater than 50% was interpreted as high heterogeneity and greater than 75% as very high heterogeneity. Forest plots provide a graphical display of study-specific phenotypic prevalence estimates and opportunity to assess between-study heterogeneity. Funnel plots and regression tests are used to assess publication bias. Sensitivity analysis was conducted by removing studies assessed through the ROB assessment as having very high levels of bias as well as any study in which fewer than 100 samples were tested.

## 5. Conclusions

Based on point prevalence estimates published in the literature, a baseline occurrence of 4.75% of surveyed food crops are calculated to harbour either TET or 3GC resistance in Enterobacteriaceae. Subgroup analysis found variation in regional pooled prevalence estimates particularly of clinically relevant 3GC resistance in Enterobacteriaceae but did not indicate a difference in AMR presence across the pre- and post-harvest stages of the food crop value chain. Despite known limitations of meta-analyses and variation in laboratory methods our results are broadly in line with other published research on AMR prevalence in food crops. To better define the risk of consumer and occupational exposure to AMR from agriculture, further surveillance is required with harmonised methods of sampling and AMR detection. This would best be done through the inclusion of food crops in integrated national surveillance schemes.

## Figures and Tables

**Figure 1 antibiotics-11-01424-f001:**
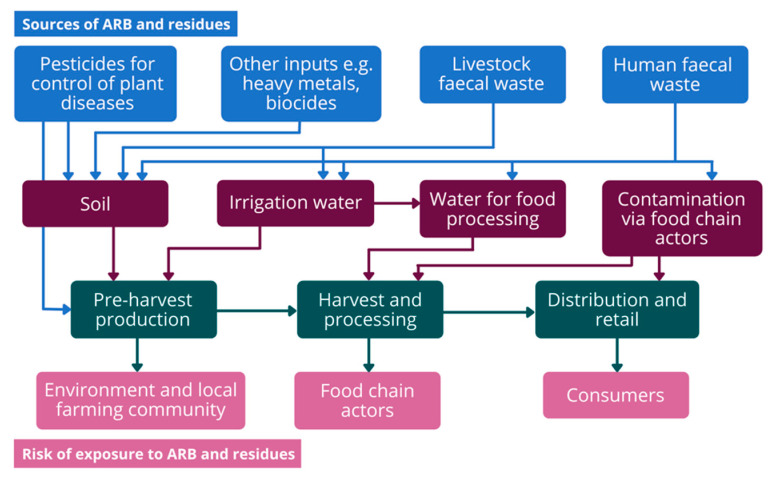
Antimicrobial resistance (AMR)-relevant pre- and post-harvest inputs in the food crop value chain.

**Figure 2 antibiotics-11-01424-f002:**
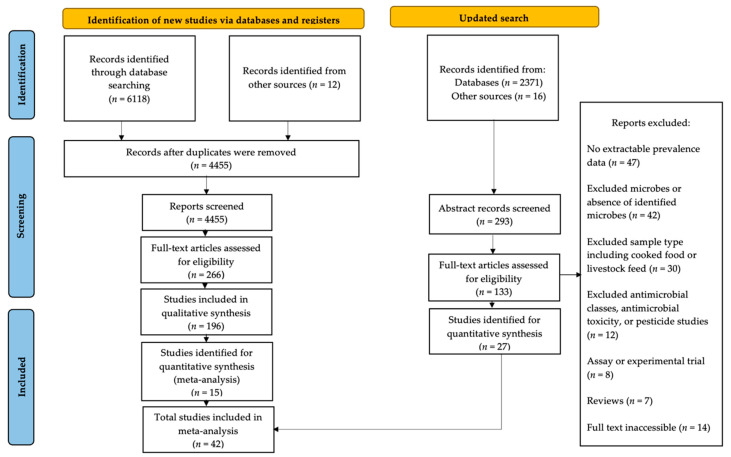
PRISMA Flow Chart: Adapted from Page et al. 2020 [32] (adapted under CC BY 4.0).

**Figure 3 antibiotics-11-01424-f003:**
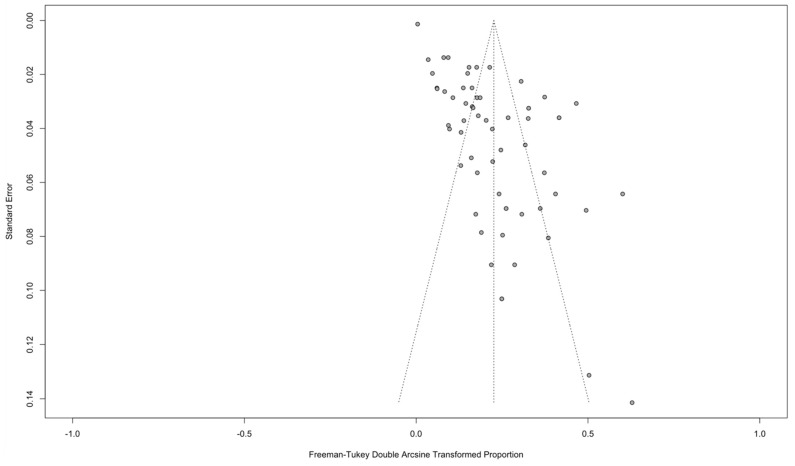
Funnel plot of included studies with phenotypic resistance.

**Table 1 antibiotics-11-01424-t001:** Risk of Bias Assessment.

Risk of Bias	Number of Studies	Citations
Very High	9	Aabed et al. 2021 [34]; Freitag et al. 2018 [35]; Hartentyo et al. 2020 [36]; Janalikova et al. 2018 [37]; Kim et al. 2015 [38]; Lopes et al. 2021 [39]; Niyomdecha et al. 2016 [40]; Pintor-Cora et al. 2021 [41]; Somda et al. 2021 [42]
High	18	Abakpa et al. 2017 [43]; Al-Kharousi et al. 2021 [44]; Ananchaipattana et al. 2014 [45]; Campos et al. 2013 [46]; Chanseyha et al. 2018 [47]; Colosi et al. 2020 [48]; Gomez-Aldapa et al. 2016 [49]; Güran et al. 2021 [50]; Kabir et al. 2014 [51]; Kholdi et al. 2021 [52]; Kurittu et al. 2021 [53]; Li et al. 2020 [54]; Najwa et al. 2015 [55]; Rasheed et al. 2014 [56]; Rodrigues et al. 2022 [57]; Romyasamit et al. 2021 [58]; Skočková et al. 2013 [59]; Zekar et al. 2017 [60]
Moderate	15	Chotinantakul et al. 2022 [61]; Haile et al. 2021 [62]; Holvoet et al. 2013 [63]; Kaesbohrer et al. 2019 [64]; Montero et al. 2021 [65]; Mwanza et al. 2021 [66]; Priyanka et al. 2021 [67]; Reddy et al. 2016 [14]; Richter et al. 2020 [68]; Shahin et al. 2019 [69]; Song et al. 2020 [70]; Sun et al. 2021 [71]; Thung et al. 2020 [72]; Van Hoeck et al. 2015 [73]; Zou et al. 2019 [74]
Low	0	None
Total studies	42	-

**Table 2 antibiotics-11-01424-t002:** Pooled prevalence estimates of Enterobacteriaceae and antimicrobial resistance in food crops.

Sub-Group	No. of Samples Tested ^a^	No. of Resistant Samples ^b^	Prevalence (%)	Prevalence CI (%)	*p*-Value	No. of Prevalence Estimates	I^2^ (%)	I^2^ CI (%)
Pathogen Prevalence								
Total Enterobacteriaceae	149,751	950	11.45	7.96, 15.45	<0.0001	44	98.5	98.3, 98.7
Pre-Harvest Samples	1442	186	13.86	5.78, 24.53	<0.0001	8	96.0	93.9, 97.3
Post-Harvest Samples	148,309	764	10.85	7.30, 14.97	<0.0001	36	98.4	98.2, 98.6
AMR Prevalence								
Total AMR	149,298	404	4.75	2.92, 6.94	<0.0001	45	97.3	96.8, 97.7
3GC Resistance								
All Stages	7841	257	3.75	2.13, 5.74	<0.0001	27	93.2	91.2, 94.7
Pre-Harvest Samples	569	33	4.45	1.44, 8.71	0.0031	5	74.9	38.2, 89.8
Post-Harvest Samples	7155	213	3.44	1.64, 5.76	<0.0001	21	94.0	92.1, 95.5
TET Resistance								
All Stages	146,037	259	4.63	2.57, 7.18	<0.0001	30	97.2	96.7, 97.7
Pre-Harvest Samples	1442	64	4.55	1.83, 8.24	<0.0001	8	86.9	76.3, 92.7
Post-Harvest Samples	144,595	195	4.72	2.11, 8.16	<0.0001	22	97.4	96.7, 97.9

^a^ The total number of plant-food samples selected for antimicrobial susceptibility testing. ^b^ the total number of plant-food samples that were reported resistant to tetracycline or third-generation cephalosporin antimicrobials.

**Table 3 antibiotics-11-01424-t003:** Pooled prevalence estimates of antimicrobial resistance in food crops sub-grouped by antimicrobial class and world region.

Region	3GC Resistance					TET Resistance					Combined AMR
	No. of Samples Tested ^a^	No. of Resistant Samples ^b^	Prevalence (CI)	*p*-Value	No. of Prevalence Estimates	No. of Samples Tested ^a^	No. of Resistant Samples ^b^	Prevalence (CI)	*p*-Value	No. of Prevalence Estimates	Prevalence (CI)
Africa	842	81	0.0659 (0.0241; 0.1240)	<0.0001	6	949	33	0.0431 (0.0087; 0.0981)	<0.0001	5	0.0544 (0.0254; 0.0924)
Americas	*	-	-	-	-	139,638	33	0.0248 (0.0000; 0.0882)	<0.0001	4	0.0420 (0.0074; 0.0988)
Eastern Med.	*	-	-	-	-	*	-	-	-	-	0.0146 (0.0036, 0.0313)
Europe	1977	20	0.0184 (0.0000; 0.0602)	<0.0001	5	807	32	0.0628 (0.0068; 0.1572)	<0.0001	6	0.0362 (0.0077; 0.0803)
South East Asia	1179	33	0.0404 (0.0040; 0.1013)	0.0293	4	1273	43	0.0509 (0.0091; 0.1177)	<0.0001	5	0.0463 (0.0163; 0.0875)
Western Pacific	2018	43	0.0278 (0.0071; 0.0586)	<0.0001	7	2281	92	0.0623 (0.0079; 0.1550)	<0.0001	7	0.0440 (0.0149; 0.0852)

* No meta-analysed results calculated where the number of prevalence estimates per sub-group was fewer than four. ^a^ The total number of plant-food samples selected for antimicrobial susceptibility testing. ^b^ the total number of plant-food samples that were reported resistant to tetracycline or third-generation cephalosporin antimicrobials.

## Data Availability

The data presented in this study are available in the Appendix A section.

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
