# Peer review of "A Meta-Analysis to Estimate Prevalence of Resistance to Tetracyclines and Third Generation Cephalosporins in Enterobacteriaceae Isolated from Food Crops"

_antibiotics, 2022, doi:10.3390/antibiotics11101424_

Round 1

Reviewer 1 Report

Dear authors,

Congratulations for your article. It seems me very interesting and it could be useful as a base or help for the One Health program. Little is known about the AMR in food crops.

In my opinion, this article is ready to be published, but with modifications, many of them in tables and supplementary material. I enclose the article, in PDF, with comments. Please, read them carefully. On the other hand, I have some comments about the Supplementary material, described below:

- Figure S1: Authors should explain briefly the results given in the table, including the colours, in order to better understand the graphic.

- Table S1: What do the authors mean? Citations or studies used in this article? On the other hand, authors must check the table and organise it, because some rows are confused with the above/below ones.

- Table S1: Microbial names always in italics and well written, including "Salmonella" in the same row, please.

Author Response

Thank you for your comments on our meta-analysis. We appreciate the opportunity to improve our manuscript and our responses to your comments are below.

On line 94, we have changed the format of the six-digit number to include a comma, we have also made this change to the numbering throughout the manuscript including in all tables and supplementary materials containing any numbers with more than three digits. We hope this improves the reading of the numbers.

We also increased the font size for Figure 2 and hope this is now more readable.

In Table 2, line 127, we changed the column titles from 'No. of events' to 'No. of resistant samples' and included a footnote line 136 to explain the column contents. We also improved the column title for 'No. of samples' to 'No. of samples tested' with further explanation from a footnote (line 135). The same changes were made to Table 3 (lines 178 - 179) and an explanation of the term 'events' which also occurs in Suppl. Figure S1 was provided in a textbox below the figure.

Consistency of font bolding for section titles occurred at line 215, 343, and 408.

Thank you also for alerting us to the lack of an explanation about the difference in time period between the SR literature search (2000 - 2022) and the meta-analysis (2010 - 2022). We have provided an explanation for this from line 363 - 369 which we hope clarifies our efforts to reduce heterogeneity in our data and improve the comparability of results across studies.

As per your suggestion, we also removed the author contributions at line 343, since they are included already in the Author Contribution section at the end of the manuscript.

We have also updated the supplementary materials to include:

1) A textbox with explanation of Figure S1, including the colouring.

2) Revised the supplementary materials Table S1 so that the included studies follow the order of citations in the Forest Plot.

3) Italicized microbial names in Table S1.

Thank you again for your comments. We hope that we have responded to everything sufficiently and believe that this feedback has improved the quality of our manuscript.

Reviewer 2 Report

The paper can be accepted for publication in its present form.

Author Response

Thank you for your feedback, it is greatly appreciated.

Reviewer 3 Report

The manuscript by Brunn et al. is well written with an original and excellent contribution to the understanding of the resistance to tetracyclines and third generation cephalosporins in Enterobacteriaceae isolated from food crops at worldwide level. The approach is very complex with a critical analysis of the presented data suggesting of the last explored topics in the field able to point up significative findings.

I suggest,  two minor point for the improvements:

Lines 32, 92, etc. - „we focused”, „we retrieved” – please avoid the using of personal mode verb formulations, it is not so characteristic to the scientific style. Please revise this issue throughout the manuscript!

Please uniformly bold the publication year for each article from the reference list.

Author Response

Thank you for this feedback on our study. We have noted the occurrence of sentences written in the first person format (using "we") and amended these to an objective format (e.g. see line 303, "this study aimed to...") or exchanged the term to "the authors" (e.g. see line 216, "the authors contend..."). Please see the revised manuscript with tracked changes throughout with our changes to this style of writing.

We have also revised our reference list so the publication year is uniformly in bold throughout.

Thank you for your feedback which helps to improve our manuscript.